# Statistical Analysis of the Long-Term Influence of COVID-19 on Waste Generation—A Case Study of Castellón in Spain

**DOI:** 10.3390/ijerph19106071

**Published:** 2022-05-17

**Authors:** Miguel-Ángel Artacho-Ramírez, Héctor Moreno-Solaz, Vanesa G. Lo-Iacono-Ferreira, Víctor-Andrés Cloquell-Ballester

**Affiliations:** 1Project Management, Innovation and Sustainability Research Center (PRINS), Universitat Politècnica de València, 46022 Valencia, Spain; miarra@dpi.upv.es (M.-Á.A.-R.); valoia@upv.es (V.G.L.-I.-F.); vacloque@dpi.upv.es (V.-A.C.-B.); 2Ayuntamiento de Castelló de la Plana, 12001 Castelló de la Plana, Spain

**Keywords:** COVID-19, waste generation, inferential analysis, long term effects

## Abstract

Existing research recognizes the COVID-19 impact on waste generation. However, the preliminary studies were made at an early pandemic stage, focused on the household waste fraction, and employed descriptive statistics that lacked statistical support. This study tries to fill this gap by providing a reliable statistical analysis setting inferential confidence in the waste generation differences found in Castellón. Repeated measures ANOVA were carried out for all the waste fractions collected and recorded in the city landfill database from 2017 to 2020. Additionally, Bonferroni’s multiple comparison test (*p* < 0.05) was used to assure confidence level correction and identify which pairs of years’ differences appeared. The longitudinal study identified trends for each waste fraction before the pandemic and showed how they changed with the advent of the crisis. Compared to 2019, waste collection in 2020 significantly grew for glass and packaging; remained unchanged for beaches, paper and cardboard, and dropped substantially for households, streets, markets, bulky waste, hospitals, and recycling centres. Total waste showed no differences between 2017 and 2019 but dropped significantly in 2020. These findings may help us better understand the long-term implications of COVID-19 and improve municipal solid waste management in a similar crisis.

## 1. Introduction

The management of municipal solid waste (MSW) has become an increasing global concern as urban populations continue to grow and consumption patterns change. MSW is one of the essential human activities with social and environmental effects [1]. For that reason, the health and environmental implications associated with MSW management are being increasingly analysed and studied [2].

Once the World Health Organization (WHO) declared the coronavirus disease of 2019 (COVID-19) a global pandemic on 11 March 2020, governments worldwide implemented measures to control the spread of the virus [3,4]. From nationwide lockdowns to access limitations or travel restrictions, various disruptive measures seriously affected global supply chains, industries, services, and financial markets—and there was an unprecedented impact on the economy, environment, and people’s lives [5,6]. People had to change the way they worked, studied, and interacted, abruptly modifying habits and the methods in which they consumed products and services [7,8,9].

The COVID-19 crisis was expected to influence waste generation, given the relationship that MSW has with socioeconomic changes, behaviour, and lifestyle [10,11,12,13]. A considerable amount of literature has been published recently analysing COVID-19’s impact on waste generation in several cities and countries. However, most of these preliminary studies were made at an early stage of the pandemic when the situation was still developing, leading to controversial results in the amount of waste generated compared with a pre-pandemic scenario. For example, Charlebois et al. [14] and Zand and Heir [15] found an increase in food waste in Canada and Tehran, respectively, and Principato et al. [16] and Fan et al. [17] found the opposite in Italy and Shanghai, respectively. These pioneering studies, with some exceptions [3,18], gathered data during the lockdown for short periods (a few months) and were focused on the household fraction [19,20,21,22].

Moreover, these studies employed interviews, focus groups [23], secondary data [24], and surveys without reaching representative sample sizes and non-probabilistic sampling methods, thus affecting the representativeness of the findings. Additionally, up to date studies, with some exceptions [3], are focused on measuring the amount of waste [25] without considering changes in waste fractions and composition, which is crucial for analysing the influence of COVID-19 on waste management [17]. Thus, despite the fact that these pioneering works provided a preliminary and valuable understanding of the potential impacts of COVID-19 on waste collection and disposal behaviour, many were descriptive in nature and lacked statistical support. Therefore, the generalisability of their findings is limited. Table 1 provides a summary of these preliminary works, showing the mentioned limitations, e.g., their focus on short periods (lockdowns), their reliance on self-reported and indirect data, the prevalence of the analysis of the household waste fraction, and the derivation of results and conclusions from descriptive data analyses. It is worth noting that information about statistical analyses in Table 1 only refers to the methods used in previous works to analyse differences in the waste generation between the pre-COVID-19 and COVID-19 scenarios.

The present work tries to fill this gap by providing a reliable statistical analysis setting inferential confidence in the waste generation differences found in Castellón, a medium-sized city in the Valencia region of Spain. Specifically, the authors compared waste generation in the city from 2017 to 2020, considering all waste fractions as recorded in municipal landfill data. Thus, the starting hypothesis is statistically significant differences in the waste generation between pre-COVID-19 and COVID-19 scenarios in the long term. These differences will depend on the analysed fractions.

The remainder of this paper is structured as follows: Section 2 is concerned with the case study description and the used methods for data analysis in this study, Section 3 presents and discusses the obtained results for each one of the waste fractions analysed, and finally, Section 4 provides the conclusions drawn from findings, the impact they may have on MSW management, the limitations of the study and further work to be carried out.

## 2. Materials and Methods

### 2.1. Case Study

The research was undertaken in Castellón. The city’s population increased 2.7% between 2017 and 2020 and reached 174,262 inhabitants [37]. Castellón is not a tourist town. It has a negative floating population of about 15,000 inhabitants who move to other tourist destinations during July and August. In the distribution of companies according to activity, the services sector, in which the HORECA (hotel, restaurant, and catering) channel predominates, is the most important (68%), followed by construction (21%) and, finally, industry (11%) [37].

The city was in lockdown along with all of Spain from 15 March until 21 June 2020. All non-essential establishments were closed, such as cafes, restaurants, hotels, and commercial and retail businesses, but internet commerce and catering services continued to operate. Food stores and supermarkets, considered essential establishments, were open while open-air markets were closed. Regarding health centres and hospitals, consultations were made by telephone, with visits to health centres only allowed when hospitalisation was necessary or in other exceptional cases [38,39].

On 28 April 2020, a national plan for asymmetric de-escalation began. Some establishments (such as hospitals, restaurants, and coffee shops) opened with some commercial shops with a restricted capacity [40]. On 21 June 2020, the state of alarm ended, and Spain entered a situation called the “new normal”. A capacity limit of 75% was generally enforced in all spaces, both outdoors and indoors, including markets, beaches, and pools [41]. The Spanish government decreed the second state of alarm on 25 October 2020, established a curfew between 12:00 a.m. and 6:00 a.m., and announced restrictions on travel between regions [42]. Measures changed during this period. From 21 December 2020, establishments were allowed to accommodate larger groups of people and stay open until 11:00 p.m. The second state of alarm ended on 9 May 2021 [43].

Figure 1 shows the six districts of Castellón. Between the port district and the main nucleus of Castellón, there is the Marjalería area with dispersed single-family houses.

A similar urban composition can be found around the periphery. The industrial sector is concentrated around the urban nucleus, except for the service sector that is within. From a waste manager’s point of view, the service is highly heterogeneous and must cover nearly four kilometres of beach managed by the municipality.

Waste managed by Castellón falls into two categories: Containerized, which includes waste deposited in containers on public roads or in public places, e.g., markets, hospitals, etc., and non-containerized which is waste from cleaning services or special wastes, e.g., beach cleaning, street cleaning, bulky waste, etc. This fraction is commonly called “Other MSW”.

All waste is categorised as MSW, including all origins (HORECA channel establishments, markets, hospitals, and industry), providing it meets the definition set by the Spanish Act 22/2011 [45].

Containerized waste is categorised as recyclable or non-recyclable (mixed). Recyclable waste refers to waste separated from the waste stream and set aside for purposes of recovery, reuse, or recycling. Mixed MSW includes urban garbage (domestic and HORECA), itinerant markets, the hospital fraction that can be assimilated to urban waste, and daily collections from grey containers. This waste is destined for a two-phase treatment plant with a semi-automatic mechanical separation and a composting tunnel [46]. The recyclable fraction is divided into four categories by colour and delivered to each corresponding treatment plant for management. Table 2 shows the frequency and characteristics of the analysed waste fractions in this study. It should be noted that bio-waste was implemented in September 2020, so it is not included in this analysis.

In addition to these fractions, other types of waste such as batteries, asbestos, tires, or construction and demolition waste are collected by special collection services.

### 2.2. Data Collection and Statistical Analysis

Information was gathered from a system connected to the scales used by the collector trucks for each fraction. The total waste per year was calculated as the sum of all the waste fractions analysed. The period under analysis is from 1 January 2017 to 31 December 2020 for all six city districts. Waste collected in 2021 was excluded because bio-waste was included as a new waste fraction in the city, thus altering the amount of waste collected in other waste fractions and avoiding comparability with previous years.

A repeated-measures analysis of variance (RM ANOVA) with the waste fractions as dependent variables was conducted to analyse if statistically significant differences in the mean for waste appeared over the years. RM ANOVA can be used for investigating changes in mean scores over three or more time points [47]. Thus, the collection of all waste fractions for 2017, 2018, 2019 and 2020 were compared with significant levels of differences set at *p* < 0.05. Data were explored to identify and exclude outliers. To test if data are normally distributed, the Kolmogorov-Smirnov test and the Shapiro-Wilk test were used for waste fractions recorded daily (>50 cases) and for waste fractions recorded every month (<50 cases), respectively [48]. Mauchly’s criterion test was used to assess if the covariance structure satisfies the sphericity condition [47]. If the sphericity assumption was violated, a Greenhouse-Geisser correction was used. Bonferroni’s multiple comparison test (*p* < 0.05) was used to assure confidence level correction and identify between the pair of years in which the differences appeared. The Eta squared value (η^2^) was calculated to measure the effect size, setting the size of the differences found (η^2^ = SS_effect_/SS_total_, where: SS_effect_ is the sum of squares for the effect that is being studied, and SS_total_ is the total sum of squares for all effects, errors, and interactions in the RM ANOVA study).

All data analyses were performed using the SPSS 16 statistical application for Windows.

## 3. Results and Discussion

Outliers appeared in all the waste fractions recorded daily for at least one year (households, hospitals, markets, and streets). After excluding those outliers, all the waste fractions analysed were normally distributed (*p* > 0.05 in all the normality tests performed). RM ANOVAs carried out for all the waste fractions show significant differences between years. The significance of Mauchly’s test, F values, significance, and η^2^ values are shown in Table 3. Mauchly’s test of sphericity for hospital, beach, and glass waste indicated that the assumption of sphericity had been violated (Mauchly’s test (sig.) < 0.05) and, therefore, a Greenhouse-Geisser correction was used.

Waste fractions were classified into three groups according to the number of statistically significant differences found in their means among the years. Group 1 is defined by fractions of waste in which statistically significant differences appeared for all pairwise comparisons (see Figure 2, and Table 4). In Group 1, a statistically significant increasing trend is observed for packaging collection from 2017 to 2020. In the case of markets and bulky waste, the increasing trend stops with a significant decrease in the amount of waste collected in 2020. In the case of the recycling centre bulky waste, the increasing trend stopped in 2018 with a significant decrease in the amount of waste collected in 2019 and in 2020. Implementing the scheduled “door-to-door” collection service for the bulky waste in 2019 could explain both the increase in bulky waste and the significant decrease in the bulky waste from recycling centres.

The environmental awareness campaigns conducted by administrations and non-governmental organisations in the years preceding the COVID-19 pandemic have led to a steadily growing collection of recyclable waste fractions (packaging, glass, and paper). However, the results in the packaging fraction in 2020 should be explained by the plastic waste boom during the pandemic. Many factors contributed to this increase in plastic waste. The pandemic induced impulsive and irrational stockpiling of groceries and essential plastic packaged products [49]. A loss of faith in unpackaged products in a new hyper hygienic approach significantly increased the use of single-use plastic (SUP) [50]. The temporary relaxation of bans on SUP bags in supermarkets also contributed to this trend [51,52]. Moreover, lockdowns and quarantines increased online food delivery and e-shopping, increasing the use of plastic-based packing materials [53,54]. Finally, the remarkable increase in the use of masks, gloves, protective suits, hand sanitizer bottles, and all kinds of personal protection equipment (PPE), along with the increase in pharmaceutical packaging waste, led to more plastic packaging waste [55,56,57]. This increase in plastic pollution poses new challenges for effective plastic waste management that need to be addressed with new strategies and directives [50,58].

In the specific route of the municipal markets, the trend for waste had been upward in recent years, except for 2020, with a significant drop in the average daily collection. This is partly due to the decline in economic and commercial activity and the protocols established by the health authorities regarding the disinfection of posts and other measures that were difficult to comply with and that led to the temporary closure of many market stalls. The drop in collection could also be related to the increase in the packaging waste fraction. The loss of faith in unpackaged products could partly explain this, along with the increased buying of less-perishable products [49,59], the closure of restaurants, and the increase in online food delivery forced by lockdowns. The wish to avoid densely populated places led people to buy food and groceries from smaller nearby outlets, limiting the purchase time spent, and fewer family members were involved in offline grocery shopping [60], which may have played a role in reducing activity in municipal markets.

Bulky waste collection in 2020 may be related to lockdowns and a drop in domestic refurbishments and economic activity. Lockdowns and mobility restrictions can also be responsible for the decrease in bulky waste taken to recycling centres by the public.

Group 2 contains fractions of waste in which no differences were found for a few years (Figure 3, Table 5). The glass fraction of waste showed no differences between 2017 and 2018 but grew significantly in 2019 and 2020. For household, hospitals, and recycling centre pruning, waste grew and showed statistically significant differences between 2017 and 2018. Such a difference disappeared between 2018 and 2019. Finally, waste decreased significantly in 2020. Finally, both paper and cardboard and beaches showed no differences in mean waste between 2019 and 2020, respectively, putting an end to a steadily growing and steadily decreasing trend in previous years.

The remarkable increase of glass collected in 2020 is in line with the findings of Filho et al. [29]. This increase could partly be explained by the decline in activity in hotels and restaurants. Lockdowns and reduced activity due to indoor capacity restrictions meant glass consumed at restaurants and hotels that used to be repackaged and sent back to beverage companies was consumed at home and placed in glass waste containers. Moreover, according to Tchetchik et al. [9], the COVID-19 crisis made people more prone to increase recycling and further reduce consumption. The authors found that the perceived link between exposure to the pandemic threat and climate change and economic vulnerability increased pro-environmental behaviour. An increase in recycling at home could also explain the increased volumes of paper and cardboard, although with no significant differences compared to 2019. The slowdown in economic activity for the leading paper and cardboard producers like shops, the hospitality industry, and industrial sectors [61] could explain the reduced volume of paper and cardboard waste in 2020.

The household fraction shows a statistically significant decrease in 2020. As said in the introduction, previous studies analysing the influence of pandemics on household waste led to controversial results. The descriptive nature of these studies mainly focused on lockdown periods, which made it difficult to evaluate such an influence in the long term. This work enables us to state that household waste volume in a pandemic was significantly lower than in pre-pandemic years. These results could be related to a generalised decrease in family consumption from the economic contraction provoked by the pandemic. Moreover, several authors found that changes to family routines (e.g., working from home, better time management, more organised purchase and cooking habits, greater use of ‘smart food delivery’) due to the pandemic may have also played a role in the observed reduction in household waste [21,36,62].

The hospital fraction also showed a statistically significant decrease in 2020, which is in line with the observed household trend. This parallel trend could be because the waste from hospitals managed by city councils can be assimilated into urban waste and does not pose any special management requirements since it was only used by non-infectious patients. It should be noted that hospital waste that is not assimilable to domestic waste (hazardous waste) which used to represent 15% of waste in healthcare facilities [63] is not collected through municipal services and has a specific channel for its management. Hazardous waste in Castellón probably grew exponentially, as happened in other cities such as Wuhan (0.6 kg/patient to 2.5 kg/patient) [64] or countries like Jordan (3.95 kg/patient to 14.16 kg/patient) [65], and this should be analysed in future studies because the pandemic has created a huge burden of healthcare systems that must also treat and properly dispose of all the waste that could further spread the SARS-CoV-2 virus [66].

Regarding recycling centre pruning waste, the reduction in 2020 could be again related to mobility restrictions which may have hampered disposals at recycling facilities. In the case of beach waste, the collection includes vegetable matter washed ashore and the elimination of algae in the sand, which has been growing in recent years due to an increase in sea temperatures. A new collection procedure set in 2018 led to a remarkable decrease in waste. Since then, the beach material removed is accumulated in points far from the shore, where it is allowed to dry to reduce the moisture content, leading to a weight reduction in the waste to be managed. Once the material has dried, it is transported to a screening plant to recover the maximum amount of sand for spreading again on the beach, thereby significantly reducing the amount of final waste. Finally, following the Ministry of Ecological Transition recommendations to enhance the circular economy, part of the algae is buried in the beach dunes to reinforce and regenerate them. These algae serve as an organic substrate that contributes to their recovery. No significant differences appeared between 2019 and 2020 despite cleaner beaches being reported as one of the positive side effects of COVID-19 on the environment [67].

Finally, Group 3 contains fractions of waste whose collection values showed no significant differences in mean values over four years. These waste fractions were streets and total waste, in which no differences appeared between 2017 and 2019, whereas the mean value significantly dropped in 2020 (see Figure 4 and Table 6).

The significant drop in street waste collected from public roads could be related to the lockdowns, mobility restrictions, and the closure of the hotel, restaurant, and nightlife industries. Total waste shows the same trend as streets. Some preliminary studies predicted an increase in MSW due to the COVID-19 pandemic, and may be influenced by the initial panic buying experienced worldwide [68,69]. However, the results obtained considering one year for the city of Castellón (with statistical support) show the opposite. These results are in line with Cai et al. [18]. These authors found that compared to normal periods in 2019, significant decreases in total waste were observed in most of the months in 2020 in Montreal (e.g., by 9.5% in May 2020) and Trento (e.g., by 13.7%, 25.3%, 14.7% from March to May 2020 and 16.5% in January 2021). Thus, it could be said that the shutdown of most of the productive and commercial activities due to the COVID-19 crisis resulted in significant economic losses which, in turn, led to a waste decrease in the long term.

Considering only 2019 and 2020, waste collection for 2020 significantly grew for glass and packaging; it remained unchanged for beaches and paper and cardboard, and significantly decreased for households, streets, markets, hospitals, recycling centre pruning, recycling centre bulky waste, and total waste.

## 4. Conclusions

This study sheds further light on the influence of the COVID-19 crisis on the generation of MSW. It complements the abundant previous descriptive studies in the literature with a longitudinal study based on the statistical inference that enables the quantitative establishment of the long-term impact of the pandemic on the generation of the waste fractions collected by the municipality of Castellón.

The comparison performed on waste fractions from 2017 to 2020 using RM ANOVA enables establishing waste trends and how these vary because of the effect of the pandemic. The results show that waste collection patterns significantly changed in Castellón city in 2020 because of the impact of COVID-19. Thus, the starting hypothesis has been confirmed, as statistically significant changes in waste generation appeared, and the analysed waste fractions were affected in different ways. Considering only 2019 and 2020, waste in 2020 significantly grew for glass and packaging; remained unchanged for beaches and paper and cardboard; and significantly dropped for households, streets, markets, hospitals, recycling centre pruning, and recycling centre bulky. Finally, total waste showed no differences from 2017 to 2019 but significantly dropped in 2020.

The start of the COVID-19 pandemic provoked changes in the waste amount, composition and distribution, safety and infection risk, and disposal rate, increasing the complexity of waste management worldwide. Changes in waste generation resulted in storage, transportation, disposal, and treatment challenges. The response of public authorities and municipal waste operators in Castellón was to quickly adapt their waste management systems and procedures to the new scenario. Ensuring the safety of the staff was a priority, followed by guaranteeing collection services with the same frequency as usual, including on-demand collection. Municipal waste collection staff were provided with additional personal protective equipment (PPE) and trained about safety measures and new protocols for disinfection equipment and vehicles. Differences in waste generation among fractions forced MSW managers to reorganise resources and adapt routes. For example, the increase in packaging and glass fractions in Castellón led to increasing collection frequencies to avoid container overflow. Given the reduction of the street waste fraction, the scheduled operations in streets were reduced and reorganized to improve wet cleaning services, such as sweeping and disinfection in markets, hospitals, and public roads, cleaning litter bins, and washing containers. The pandemic crisis mainly affected collection systems, but it progressively reached other players such as recyclers in the long term. Sorting and treatment systems also experienced some disruptions because new restrictions appeared for manual sorting and recycling due to safety precautions. Daily waste operations were also affected by increased monitoring to avoid illegal dumping, shortages of personnel, and increased communication to inform citizens about adequately managing their waste. As can be seen, despite the fact that the total waste dropped, the issues to be considered by MSW managers significantly increased in number and difficulty. Despite the agile adaptation of the administration and MSW managers, all the changes were made reactively. If they had had the objective and reliable data about changes in material flows obtained in this study, they could have planned their actions better, reorganizing resources more efficiently. Additionally, the present paper’s findings might better help understand the long-term implications of COVID-19 and prepare planners and policymakers for changes in the waste stream due to pandemics or other unprecedented emergencies.

However, a note of caution is due here since previous studies showed that waste generation and composition might vary depending on the location [70], socioeconomic factors, or climatic factors [10,12,71,72]. Thus, further studies should be carried out in cities of different sizes and with different economic activities. Such studies should include the analysis of hazardous medical waste not analysed in the present study. Moreover, the pandemic persists, so further studies should analyse whether the results remain visible, return to the previous trend, or whether there is a rebound effect after returning to normality. Thus, long-term analyses of the total impact of COVID-19 on resources and waste management and the dynamics of material flow seem necessary. Finally, waste management and planning practices applied after the advent of the pandemic should be evaluated with the goal of identifying managerial improvements.

## Figures and Tables

**Figure 1 ijerph-19-06071-f001:**
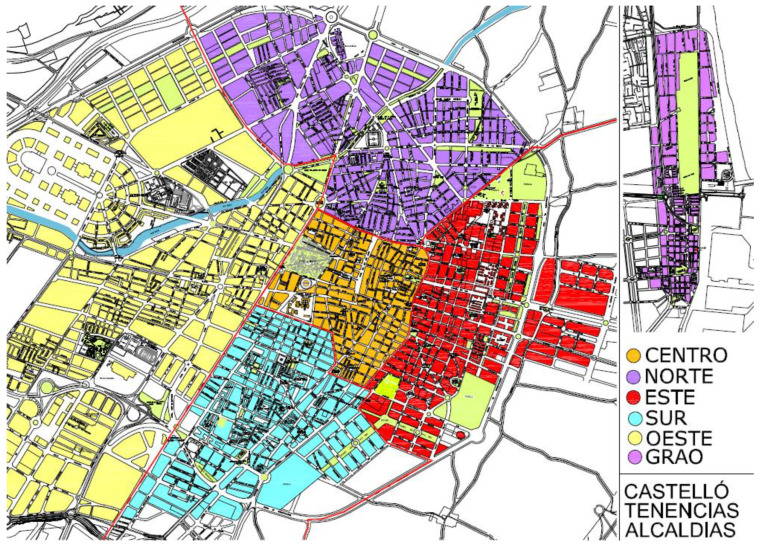
Castellón map with districts (39°59.1402′ N, 0°2.961′ W). Adapted with permission from Ayuntamiento de Castelló de la Plana [44].

**Figure 2 ijerph-19-06071-f002:**
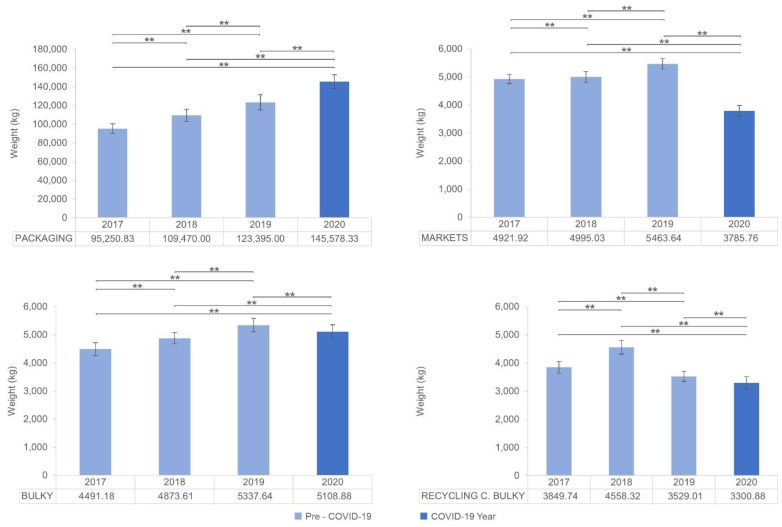
Mean and 95% CI for each year in waste fractions of Group 1: packaging, markets, bulky and recycling centre bulky waste collection. (**) for *p* < 0.01.

**Figure 3 ijerph-19-06071-f003:**
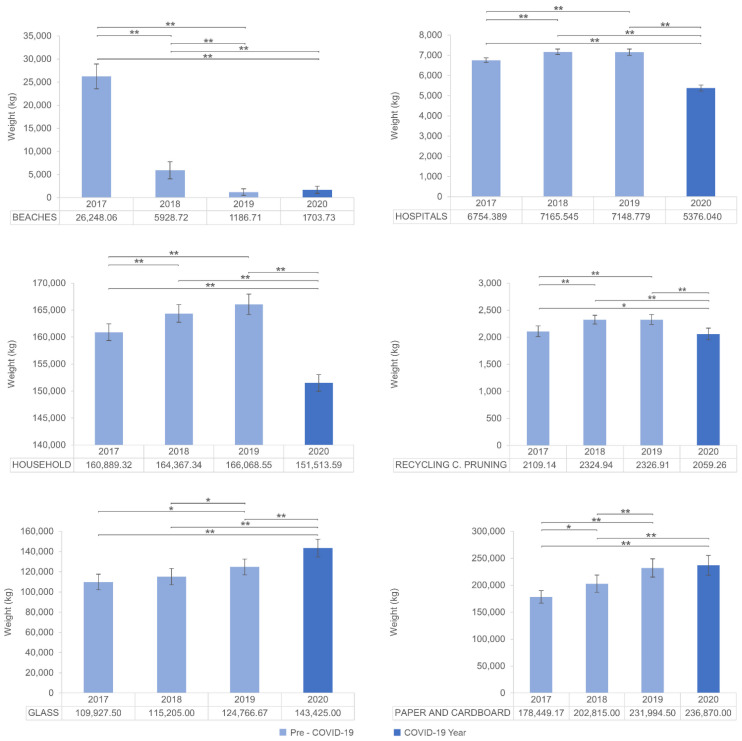
Mean and 95% CI for each year in waste fractions of Group 2: glass, household, hospitals, recycling centre pruning, paper and cardboard and beaches waste collection. (*) is used for *p* < 0.05 and (**) for *p* < 0.01.

**Figure 4 ijerph-19-06071-f004:**
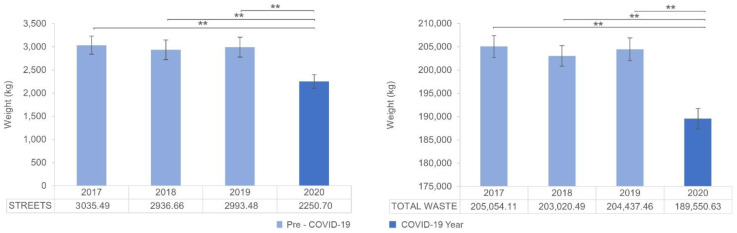
Mean and 95% CI for each year in waste fractions of Group 3: streets and total waste collection. (**) for *p* < 0.01.

**Table 1 ijerph-19-06071-t001:** Summary of preliminary works on waste generation differences found between pre-COVID-19 and COVID-19 scenarios.

Work	Country	Period Analysed	Methodology	Waste Fractions	Statistical Analysis
Amicarelli & Bux [21]	Italy	March to May 2020	Food diaries	Household	Descriptive (sum of reported data)
Charlebois et al. [14]	Canada	August 2020	Survey	Household	Descriptive (percentages)
NZWC & LFHW [26]	Canada	June 2020	Survey	Household	Descriptive (percentages)
Principato et al. [16]	Italy	March to April 2020	Survey	Household	Descriptive (percentages)
Bogevska et al. [27]	North Macedonia	May to June 2020	Survey	Household	Descriptive (percentages)
Aldaco et al. [24]	Spain	March to April 2020	Secondary data	Household	Descriptive (Difference)
Jribi et al. [19]	Tunisia	March to April 2020	Survey	Household	Descriptive (percentages)
Ben Hassen et al. [28]	Qatar	May to June 2020	Survey	Household	Descriptive (means, variation ratio, frequencies, and percentages)
Ismail et al. [20]	Malaysia	March to April 2020	Secondary data	Household	Descriptive and One-way ANOVA
Brizi & Biraglia [22]	India & USA	Lockdowns	Survey	Household	Descriptive and Sequential Mediation Model
Richter et al. [8]	Canada	March to September 2020	Landfill database	Household/Total Waste	Descriptive (measures of central tendency)
Ikiz et al. [23]	Canada	May 2020	Interviews & focus group	Household	Qualitative analysis
Zand & Heir [15]	Iran	Not reported	Estimation	Total Waste/Medical Waste	Descriptive
Fan et al. [17]	China, Singapore, Czech Republic	March to May 2020	Survey and Secondary Data	Plastic/Household	Descriptive
Cai et al. [18]	USA, Brazil, Canada, UK, France, and Italy	2019, 2020, 2021	Secondary data	Total Waste	Descriptive (variation ratio)
Richter et al. [3]	Canada	January 2018 to September 2020	Landfill database	Solid/Mixed Solid/Construction/Grit/Mixes Asphalt Shingles/Biomedical	Descriptive (boxplots)
Filho et al. [29]	41 countries	Lockdowns	Survey	Plastic	Descriptive
Olatayo et al. [30]	South Africa	2019 to 2020	Estimations from secondary data	Plastic (PPE)	Descriptive (Material Flow Analysis)
Babbitt et al. [31]	USA	March to July 2020	Survey	Household	Descriptive
Urban & Nakada [32]	Brazil	2010 to 2020	Secondary data	Household/Recyclables/Streets	Descriptive (means)
Vittuary et al. [33]	Italy	May 2020	Survey	Household (food)	Descriptive (percentages)
Strotman et al. [34]	Germany	October 2020	Survey	Household (Food)	Descriptive (percentages)
Kasim et al. [35]	Guyana and Nigenria	Lockdowns	Survey/Interviews	Household (Food)	Descriptive
Laila et al. [36]	Canada	February to August 2020	Survey/Interviews/Audits	Household (Food)	Descriptive/Non-parametric test (Wilcoxon)

**Table 2 ijerph-19-06071-t002:** Waste fraction characteristics of Castellón.

Waste Fraction	Category (Bin Colour)	Collection Frequency	Nº Containers in Street	Waste Volume Capacity (l)	Description of Waste Fraction
Packaging	Recyclable, containerized (yellow)	Two to three times a week, depending on location	713	2,139,000	Plastic bottles and bags, metal cans, mixed packaging (e.g., aluminium and paperboard)
Glass	Recyclable, containerized (green)	Every 14 days	718	2,154,000	Bottles, tins, jars, etc. glass items.
Paper and cardboard	Recyclable, containerized (blue)	Two to three times a week, depending on location	560	1,680,000	Cardboard, paper, newspapers, labels, etc.
Household	Mixed MSW, containerized (grey)	Every day	3,379	3,379,000	Traditional waste system ‘All in one’. According to law, it should only contain waste that is non-admissible in other fractions
Hospitals	Mixed MSW, containerized (grey)	Every day	140	140,000	Domestic assimilable hospital waste (paper, food, textile, etc.)
Markets	Mixed MSW, containerized (grey)	Every day	80	80,000	Domestic assimilable market waste (paperboard, food, etc.)
Streets	Non-containerized	Every day	-	-	Waste collected from ground and street litter bins
Beaches	Non-containerized	Monday to Saturday	-	-	Waste collected from sand and beach litter bins
Recycling centre bulky	Non-containerized	On demand	-	277,000	Bulky waste (furniture, home appliances, wood, etc.) deposited in recycling centre
Recycling centre pruning	Non-containerized	On demand	-	40,000	Pruning (branches, trunks, leaves) disposed in recycling centre
Bulky waste	Non-containerized	Monday to Saturday	-	-	Bulky waste disposed in streets with (or without) previous order

Adapted with permission from Ayuntamiento de Castelló de la Plana [44].

**Table 3 ijerph-19-06071-t003:** RM ANOVA.

	Mauchly Test (sig.)	F	Sig.	η^2^
Household	0.087	74.626	0.000	0.908
Hospitals	0.029 *	3,489,815 *	0.000 *	0.923
Markets	0.092	3230.795	0.000	0.784
Streets	0.199	12.072	0.000	0.421
Beaches	0.000 *	31,600 *	0.000 *	0.276
Recycling centre bulky	0.189	1426.977	0.000	0.939
Recycling centre pruning	0.090	217.996	0.000	0.889
Bulky	0.201	201.264	0.000	0.970
Packaging	0.692	154.700	0.000	0.991
Paper and cardboard	0.828	53.844	0.000	0.947
Glass	0.027 *	37.150 *	0.000 *	0.772
Total waste	0.150	8616.609	0.000	0.961

* Greenhouse-Geisser correction.

**Table 4 ijerph-19-06071-t004:** Significant differences in packaging, markets, bulky and recycling centre bulky waste collection from Bonferroni’s comparison test.

Packaging			Markets	Bulky	Recycling C. Bulky
Years		Monthly Means Difference	Daily Means Difference	Daily Means Difference	Daily Means Difference
(I)	(J)	(I) − (J)	(I) − (J)	(I) − (J)	(I) − (J)
2020	2017	50,327.500 **	−1136.159 **	617.700 **	−548.864 **
	2018	36,108.333 **	−1209.272 **	235.272 **	−1257.436 **
	2019	22,183.333 **	−1677.881 **	−228.754 **	−228.132 **
2018	2019	−13,925.000 **	−648.609 **	−464.026 **	1029.304 **
2017	2018	−14,219.167 **	−73.113 **	−382.428 **	−708.571 **
	2019	−28,144.167 **	−541.722 **	−846.454 **	320.733 **

** *p* < 0.01.

**Table 5 ijerph-19-06071-t005:** Significant differences for glass, household, hospitals, recycling centre pruning, paper and cardboard and beach waste from Bonferroni’s comparison test.

Glass			Household	Hospitals	Recycling C. Pruning	Paper and Cardboard	Beaches
Years		Monthly Mean Differences	Daily Mean Differences	Daily Mean Differences	Daily Mean Differences	Monthly Mean Differences	Daily Mean Differences
(I)	(J)	(I) − (J)	(I) − (J)	(I) − (J)	(I) − (J)	(I) − (J)	(I) − (J)
2020	2017	33,497.500 **	−9375.726 **	−1378.350 **	−49.877 *	58,420.833 **	−24,544.327 **
	2018	28,220.000 **	−12,853.753 **	−1789.505 **	−265.654 **	34,055.000 **	−4224.993 **
2018	2019	18,658.333 **	−14,554.959 **	−1772.739 **	−267.654 **	−29,179.500 **	4742.011 **
2017	2018	−9561.667 *	−3478.027 **	−411.155 **	−215.802 **	−24,365.833 *	20,319.335 **
	2019	−14,839.167 *	−5179.233 **	−394.389 **	−217.778 **	−53,545.333 **	25,061.345 **

** *p* < 0.01. * *p* < 0.05.

**Table 6 ijerph-19-06071-t006:** Significant differences in streets and total waste collection from Bonferroni’s comparison test.

Streets			Total Waste
Years		Daily Means Difference	Daily Means Difference
(I)	(J)	(I) − (J)	(I) − (J)
2020	2017	−784.791 **	−15,503.484 **
	2018	−685.961 **	−13,469.868 **
	2019	−742.789 **	−14,886.838 **

** *p* < 0.01.

## Data Availability

Not applicable.

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
