# Peer review of "Statistical Analysis of the Long-Term Influence of COVID-19 on Waste Generation—A Case Study of Castellón in Spain"

_ijerph, 2022, doi:10.3390/ijerph19106071_

Round 1
Reviewer 1 Report
Dear authors, congratulation on completing the manuscript. My comment is as follows:
Abstract: The abstract is clearly written. However, there is room for improvement. For example, the sources of data collection are not clear.
Introduction: this section is clear. However, the justification of the study is not convincing. The authors have justified that there is a lack of longitudinal study on waste generation during the covid pandemic without any proof and analysis. My suggestion is to support this statement with some analysis.
Methodology: the authors need to justify why choosing the year 2017 until 2020. why not start in 2018 and 2019? and why not extend the data collection until 2021?. Besides that, I did not notice any null hypotheses discussed in this section since the researcher used RM ANOVA. How does the author ensure that the data is normally distributed before being analysed using RM ANOVA (Assumption of RM ANOVA)?
Results and Discussion: I think the results are hard to follow. My suggestion is to simplify the data and make it presentable. I did not say that your data is wrong, but the way it's being presented is not concise. In the discussion section, I think the authors need to discuss the result critically.
Conclusion: My suggestion is to move further study recommendations to this section. Overall, the conclusion is ok.
Reviewer 2 Report
The current work deals with the potential long-term threats of waste generation in the COVID-19 pandemic with special reference to Castellón in Spain. The work is timely and presents insightful measures that should be taken care of to avoid any future consequences due to COVID-19 waste disposal. After a careful review, I found this manuscript interesting and suitable for publication in the IJERPH journal. However, I would like to point out many grey points that should be addressed as major revisions. My specific comments are:
- Correct the grammatical and typo errors throughout the manuscript. There are many in the current version.
- The title of this manuscript includes “Statistical analysis”. However, I can not see any other test except ANOVA which is very common. I suggest including some other tests, such as PCA or clustering to derive more appealing findings and support the data using statistics actually.
- “covid-19” should be changed to ‘COVID-19” in the whole manuscript. Define its full form at first use.
- “&” should be changed to “and”.
- Line 23: What is “MSW”? Provide full form.
- The dates are missing their year in many places, e.g. lines 79, 81, …. and many more.
- Introduction: Well written.
- Materials and methods: Not fluent. Create subheadings to discuss the specific methodologies adopted in the paper.
- Check the font of all tables.
- Add geocoordinates of all study and sampling sites.
- Source of the map in Figure 1? The legends are not visible. Increase the font size.
- p<.05 should be changed to p<0.05.
- Improve the quality, color histogram, legends, and axial lines/texts. The figures look very ordinary and seem to be drawn by a kid. The font of the figure should be the same as in the manuscript text.
- Similar types of figures/tables must be merged together to reduce their number and avoid a redundant flow of reading. Table 3-14 and Figure 2-13 should be merged. Use a,b,c,d for each 4 merged figures.
- Provide the sources of data collection as a footnote to each table/figure.
- Update the references and avoid outdated ones. Add https://doi.org/10.1007/s10163-021-01281-w which is more related to the current work.
Reviewer 3 Report
Dear respected authors,
- The literature gap and the contribution of the respected authors should be mentioned clearly in the Introduction section. Just mentioning the “lacked statistical support” in line 60, is not enough for such explanations.
- It is recommended to explain the structure of the manuscript in the last paragraph of the Introduction section.
- In lines 63 and 67, the names of the city and the country of the considered case study have been repeated. It is suggested to mention it completely in line 63 and try to hesitate of such repetition immediately after each other.
- The second sentence of the first paragraph of the Materials and Methods section needs to be supported by mentioning the related reference.
- Line 97: “a waste management point of view” should be changed to “a waste manager’s point of view”.
- The second section of Materials and Methods mainly contains details related to the case study, but according to the title, the used materials and methods should be significantly explained. It is suggested to create a section/subsection entitles “Case study” to mention the details related to the events in this city/location by dates.
- In line 106, it is mentioned that “Containerised waste is categorized as recyclable, non-recyclable, or mixed”. If such classification has been taken from the literature, the related reference should be mentioned. Otherwise, it must be clear that it is considered by the authors for the first time. In the latter case, the reason for it should be mentioned too. The same issue exists for dividing the “recyclable fraction” in line 110.
- The columns of Table 1 should be mentioned clearly. Just based on the column titles, the content of them is not understandable for the potential readers. It is suggested to increase the with of the last column to decrease the length of the table.
- As “RM ANOVA” has been defined for “repeated measures analysis of variance” in line 126, it is recommended to use it in line 138 instead of the complete name of this method.
- To mention “Eta-square”, there are two different ways as shown in lines 133 and 139. It is recommended to unify it. In addition, the meaning of this character and the formula to calculate it should be mentioned in the text.
- As all the “Mean differences” have been found under p < 0.01 in Table 3, the explanation of “**p<0.01. *p<0.05” is not necessary. The same issue exists for tables 4 to 6.
- The trend of the data displayed in Figures 2 to 11 needs to be explained and discussed in detail.
- The related references are better be mentioned like “[5, 6]” in line 38, instead of “(e.g.[5, 6])”. The same issue is seen in line 42 for mentioning references [10-13], etc. Check whole the text for this issue.
- The Conclusion section needs to be extended. It should contain the aim of the study, the used methods, tools, etc. It should also contain the main results, the limitations of the study, and the future research, in separate paragraphs.
- The extended form of MSW should be mentioned in the Abstract section.
Reviewer 4 Report
The paper entitled ‘Statistical analysis of the long-term influence of covid-19 on waste generation – a case study of Castellón in Spain’ is a trial to determine the impact of covid-19 on waste collection and disposal behaviour. The subject is important in current pandemic time, but the paper needs some improvements before publication to fulfill high standards of IJERPH journal.
COMMENTS:
Line 67 – the information regarding population changes influenced by the number of tourists is needed
Line 132 – ‘(sig)’ – should be explained
All figures – add statistical significance
Figure 2 – explanation is needed of almost no changes in a linear trend in prepandemic (2017-2019) and pandemic period (2019 – 2020)
Figure 5 – explanation is needed of the reason of serious decrease in recycling center bulky between 2018 and 2019 vs small decrease in pandemic time
Figure 6 – explanation is needed of no changes (similar trend) in prepandemic and pandemic period
Line 351 – 365 – should be moved to Conclusion section
Line 366 – the Conclusion section should be extended by the sugestions coming from obtained rsults for waste management
Reviewer 5 Report
The manuscript is well-written and focused. However, the following comments need to be addressed:
- Introduction section is short and problem statement should be better emphasized in it.
- A literature review section should be separated from the introduction section to clearly describe previous research studies.
- More updated research studies need to be added to strengthen literature review section.
- It is not clear what are the shortcomings of previous research studies.
- Research Methodology section needs to be added to depict the steps of the proposed methodology by the authors.
- References need to be added for the selected statistical tests, also it is important to highlight the reasoning behind their selection.
- The authors need to apply normality tests to determine if parametric or nonparametric tests need to be applied.
- The main contributions of the present research study should be highlighted.
- Limitations and future research directions should be added at the end of conclusion section.
Round 2
Reviewer 2 Report
The authors have revised the manuscript as per my suggestions and answered all my queries well. I do not see any reason hindering the acceptance of this significantly improved manuscript version. I suggest acceptance in current form. Thank you.
Reviewer 3 Report
Dear Authors,
All the comments have been answered patiently by the authors and the manuscript has been modified precisely, accordingly. Therefore, the revised version of the manuscript is worth being published in the respected Journal of Environmental Research and Public Health.
Kind regards,
Reviewer 5 Report
The authors addressed all the comments.